# Ovarian Carcinosarcoma with Retroperitoneal Para-Aortic Lymph Node Dissemination Followed by an Unusual Postoperative Complication: A Case Report with a Brief Literature Review

**DOI:** 10.3390/diagnostics10121073

**Published:** 2020-12-11

**Authors:** Stoyan Kostov, Yavor Kornovski, Yonka Ivanova, Deyan Dzhenkov, George Stoyanov, Stanislav Stoilov, Stanislav Slavchev, Ekaterina Trendafilova, Angel Yordanov

**Affiliations:** 1Department of Gynecology, Medical University of Varna “Prof. Dr. Paraskev Stoyanov”, 9002 Varna, Bulgaria; drstoqn.kostov@gmail.com (S.K.); stoilov48@gmail.com (S.S.); 2Department of Obstetrics and Gynecology, Medical University of Varna “Prof. Dr. Paraskev Stoyanov”, 9002 Varna, Bulgaria; ykornovski@abv.bg (Y.K.); yonka.ivanova@abv.bg (Y.I.); st_slavchev@abv.bg (S.S.); doctortrendafilova@abv.bg (E.T.); 3Department of General and Clinical Pathology, Forensic Medicine and Deontology, Faculty of Medicine, Medical University of Varna “Prof. Dr. Paraskev Stoyanov”, 9002 Varna, Bulgaria; dzhenkov@mail.bg (D.D.); georgi.geesh@gmail.com (G.S.); 4Department of Gynecologic Oncology, Medical University Pleven, 5800 Pleven, Bulgaria

**Keywords:** ovarian carcinosarcoma, paraaortic lymph node metastasis, diagnosis, treatment

## Abstract

**Introduction**. Ovarian carcinosarcoma (OCS), also known as malignant mixed Müllerian tumour (MMMT), is one of the rarest histological subtypes of ovarian cancer. It is an aggressive tumour with a dismal prognosis—the median survival of patients is less than two years. The rarity of the disease generates many controversies about histogenesis, prognostic factors and treatment of OCS. Histologically, OCS is composed of an epithelial and sarcomatous component. **Case report**. In the present case, a patient with bilateral ovarian cysts and bulky paraaortic lymph nodes is reported. Retroperitoneal paraaortic lymph node metastases were the only extrapelvic dissemination of OCS. The patient underwent comprehensive surgical staging procedures, including total abdominal hysterectomy and bilateral salpingo-oophorectomy, supracolic omentectomy and selective para-aortic lymphadenectomy. Histologically the ovarian carcinosarcoma was composed of an epithelial component (high-grade serous adenocarcinoma) and three sarcomatous components (homologous—endometrial stromal cell sarcoma, and heterologous—chondrosarcoma, rhabdomyosarcoma). Immunohistochemistry staining was performed. A postoperative complication (adhesion between the abdominal aorta and terminal ileum causing obstructive ileus) that has never been reported in the medical literature occurred. **Conclusion**. Carcinosarcomas are carcinomas with epithelial–mesenchymal transition and heterologous differentiation. Retroperitoneal pelvic and paraaortic lymph nodes should be carefully inspected in patients with ovarian tumours. Adhesions between the small bowels and abdominal aorta are possible complications after lymph node dissection in the paraaortic region.

## 1. Introduction

Ovarian cancer is the fifth most common cause of cancer death among women, and ninety percent of ovarian cancers are of an epithelial cell type, while sex cord-stromal tumours and malignant ovarian germ cell tumours are relatively rare [1]. Ovarian carcinosarcoma (OCS), also known as malignant mixed Müllerian tumour (MMMT), is one of the rarest histological subtypes of ovarian cancer. OCS represents 1–4% of all ovarian tumours [2,3]. OCS can be found in the genito-urinary tract with the most common site being the uterus, followed by the ovary, fallopian tubes, cervix, vagina, peritoneum, breast and urethra [3,4]. Histologically, OCS is composed of an epithelial and sarcomatous component—homologous (i.e., having Müllerian pathology) and heterologous [2,4]. At the time of diagnosis, about half of the patients have ascites, and one in two patients will already have pelvic and para-aortic lymph node metastases [3]. OCS is an aggressive tumour with a dismal prognosis—the median survival of patients less than two years [2,5]. Prognostic factors remain controversial due to the rarity of the tumour. There is also little consensus about the most effective treatment, although optimal cytoreductive surgery and platinum-based chemotherapy appear to improve the outcomes [6,7]. In the current case study, a case of a female patient with bilateral ovarian OCS treated with optimal cytoreductive surgery is reported. An unusual postoperative complication occurred. An ethical committee approval (number 546 /30.10.2019) was obtained for it.

## 2. Case Report

A 53-year-old postmenopausal woman (gravida 1, para 1) presented at the Department of Gynecology with a short history of abdominal pain and abdominal distension. She had no relevant history of urogenital malignancy and no vaginal bleeding. She did not have a medical history of gynaecological problems, hormone therapy, or any major medical diseases—diabetes mellitus, cardiovascular, or renal issues. Gynaecological examination revealed two large palpable, relatively mobile, non-tender masses in the pelvic region. Abdominal and vaginal ultrasonography showed two ovarian tumours of maximum diameter—left tumour 12 cm, right tumour—10 cm, composed of solid and non-solid masses with mixed echogenicity—hypoechoic and hyperechoic components. There was free fluid in the pouch of Douglas. According to the International Ovarian Tumour Analysis (IOTA), bilateral ovarian tumours were classified as malignant: irregular solid tumours, presence of ascites, and very strong blood flow [8,9]. The chest X-ray showed no pleural effusion or pulmonary metastases.

Computed tomography and measurement of tumour marker levels were not performed. Laboratory data results were normal. Suspicion for ovarian malignancy prompted an exploratory midline laparotomy from above the umbilicus to the symphysis pubis [7]. During surgery, the abdominal and retroperitoneal lymph nodes were inspected precisely. Haemorrhagic ascites of 300 mL was presented in the peritoneal cavity. There were no macroscopic metastases in the liver, gall-bladder, greater and lesser omentum, alimentary organs and pelvic lymph nodes. Bulky paraaortic lymph nodes, suspected for lymph node metastases were palpated. Ascites fluid analysis was obtained. Frozen section from the left adnexal tumour revealed malignancy. The patient underwent comprehensive surgical staging, including total abdominal hysterectomy and bilateral salpingo-oophorectomy, supracolic omentectomy, and selective para-aortic lymphadenectomy. During para-aortic lymphadenectomy, the inferior mesenteric artery was ligated due to metastatic lymph node dissemination. Lesions of the inferior vena cava were also observed. Thumbtack was applied followed by ligation with an absorbable suture 4-0. Total blood loss was 500 mL (Figure 1).

The gross appearance of the ovarian tumours was soft and fleshy, with haemorrhage and necrosis. Both fallopian tubes were infiltrated, while the uterus was without macroscopic pathological changes. A histopathological examination revealed OCS in both ovaries, infiltrating both fallopian tubes. Histologically, the OCS was composed of epithelial component (high-grade serous adenocarcinoma) and three sarcomatous components (homologous—endometrial stromal cell sarcoma, and heterologous—chondrosarcoma, rhabdomyosarcoma). Immunohistochemistry (IHC) staining for cytokeratin shows diffuse intense staining of the epithelial element, while desmin staining exhibits rhabdomyosarcomatous components. Histologically, rhabdomyosarcomatous components were predominant at more than 25%. The expression of p53 protein was negative for both sarcomatous and carcinomas components. The number of para-aortic lymph nodes removed en masse during lymphadenectomy was 19—all with metastatic cancer spread from the epithelial component (high-grade serous carcinoma). Greater and lesser omentum and uterus were without metastases. Cytological findings from the ascites revealed adenocarcinoma cells. The patient was diagnosed with surgical stage IIIA1 according to the latest International Federation of Gynecology and Obstetrics (FIGO) classification, and pT2aN1bMx, according to the TNM classification [10]. On post-surgery day 10, the patient was discharged from the hospital in good health condition (Figure 2, Figure 3 and Figure 4).

On post-surgery day 15, the patient was admitted again at the Department of Gynecology with abdominal distension, abdominal pain, fever 38.2 °C, vomiting, and reduced intestinal peristalsis. Abdominal and vaginal sonography showed free fluid in the abdominal cavity. Computed tomography demonstrated postoperative ileus and a high amount of free fluid in the abdominal cavity. The patient’s laboratory results revealed hyperproteinaemia and hypoalbuminemia. The chest X ray was without abnormalities. After surgeon consultation, conservative treatment was initiated. Relaparotomy was performed as the patient’s symptoms persisted for 72 h. Abdominal wall dehiscence at the level of the umbilicus was observed. During the operation, up to 4 L of lymph fluid were evacuated from the abdominal cavity. A cytology and microbiology examination was obtained. Sigmoid and transverse colon were dilatated. Adhesion between the terminal ileum and the abdominal aorta just below the superior mesenteric artery was noticed. Adhesiolysis was achieved. Nasogastric intubation was obtained. 

Two drains were left in the abdominal cavity. The abdominal wall was closed and reinforced with synthetic mesh. Metal skin staples were also used during abdominal surgical wound closure. Postoperative empirical antibiotics—ceftriaxone (2 g every 12 h, i.v., 5 days), and metronidazole (500 mg every 8 h, i.v., 5 days), and albumin infusion therapy was initiated. On the next three postoperative days, the patient was subfebrile with normal peristalsis and diuresis. The nasogastric tube was removed. Total blood loss was 300 mL. Microbiology and cytology examinations detected no bacterial flora and no tumour cells. On post-surgery day 7, drains were removed. On post-surgery day 10, the patient was discharged without complaints (Figure 5 and Figure 6). On post-surgery day 21, metal skin staples were removed. The patient made an uneventful postoperative recovery.

## 3. Discussion

OCS is a rare tumour, which represents one of the most highly aggressive cancers of the female genital tract with poor long-term prognosis [7]. OCS is usually encountered after menopause, with a median age of 65 years [11]. Risk factors are the same as for ovarian carcinomas and sarcomas: obesity, nulliparity, exogenous oestrogen, radiotherapy and long-term Tamoxifen use [7]. Patients usually have advanced disease at the time of diagnosis. The majority of patients are Caucasian and present with stage III–IV disease and spread beyond the ovary (>90%) at the time of diagnosis [4,11,12,13]. MMMT has a worse survival rate than high-grade ovarian cancer at the same FIGO stage, as approximately 75% of reported patients die of the disease in an average of 12 months postoperatively. MMMT is also disseminated earlier than ovarian cancer [2,13]. MMMT staging is the same as for ovarian cancer [11]. In the present case, the patient was under 65 years, with no risk factors and with advanced-stage disease—FIGO IIIA1.

It is hard to distinguish MMMT and ovarian cancer preoperatively, as both malignancies have the same clinical and radiological findings [11]. The clinical presentation of MMMT is the same as the ovarian carcinoma. The most common symptoms are abdominal pain, distension, early satiety, bloating, nausea, vomiting, and weight loss [5,11,14]. The spread pattern of OCS is the same as that of ovarian carcinoma—peritoneal dissemination at the early sites of spreading [6]. Although both sarcoma and carcinoma components are seen in metastatic diseases, OCS rarely metastasizes in the lung and brain, unlike its uterine counterpart [14]. Serum CA-125 levels are elevated in the majority of cases [11].

Three theories have been suggested to explain the histogenesis of gynaecological carcinosarcoma: biclonal (collision) and monoclonal (combination and conversion) theories. The collision theory suggests that carcinoma and sarcoma are two independent tumours originating from a separate cell, which later merge. The combination theory assumes that sarcoma and carcinoma components are derived from a single stem cell and undergo divergent differentiation early in tumour evolution. The conversion theory states that an epithelial cell undergoes metaplasmic differentiation that initiates tumour genesis and gives rise to a sarcomatous component, and these tumours should be regarded as dedifferentiated carcinomas of the ovary [1,12,14]. Latest studies extrapolate that carcinosarcomas are carcinomas with epithelial–mesenchymal transition and heterologous differentiation [1,12,13,14,15]. Studies communicate that monoclonal (combination and conversion) theories are not mutually exclusive [1,12,14,15]. As OCS is a rare disease, most of the studies on the genetic origin of carcinosarcoma (CS) in gynaecologic tumours come from uterine carcinosarcoma specimens [14]. Some studies utilize the protein p53expression to explain the monoclonal theory, as OCS has been reported to overexpress p53 in a higher proportion [1,14]. The consistent concordance of p53 staining between the carcinomatous and sarcomatous elements in OCS (p53 protein expression is either negative or positive for both components) supports the monoclonal theory: if these were transverse colons, such concordance in all cases would be extremely unlikely [14,15]. Mayall et al. concluded that disparities in the presence of p53 gene mutation between the epithelial and stromal components prove the biclonal theory, whereas if carcinosarcomas were monoclonal, then disparities would not occur [16]. Although the mutation of the p53 gene does not always cause p53 immunostaining, most of the immunostained for p53 tumours have a p53 gene mutation [16]. The present case also supports the theory of common origin for the epithelial and mesenchymal elements, as the pattern of p53 immunostaining in the epithelial and stromal components of carcinosarcomas was investigated. The p53 expression was negative for both components. Other studies, which support carcinosarcoma theory, include clonality studies, genomic analysis, loss of heterozygosity, gene mutations, and expressions [1,14]. Cases describing ovarian carcinoma recurring as OCS support the conversion theory [17,18]. The three theories are demonstrated in Figure 7.

Histologically, OCS is composed of an epithelial (endometrioid, clear cell, serous, mucinous, squamous, undifferentiated), as well as of a sarcomatous component (homologous—fibrosarcoma, leiomyosarcoma, endometrial stromal sarcoma, or heterologous—osteosarcoma, rhabdomyosarcoma, liposarcoma or chondrosarcoma). Immunohistochemistry (IHC) staining for cytokeratin and epithelial membrane antigen showed diffuse intense staining of the epithelial element, while the mesenchymal component usually stains for vimentin, smooth muscle actin, desmin, and focal cytokeratin [1,19]. Positive CK7 and negative CK20 staining support Mullerian origin [1]. In the present case, cytokeratin and desmin were used for epithelial and mesenchymal components. Typical for stromal sarcoma, a starburst appearance was noticed macroscopically.

The rarity of the disease generates many controversies about the prognostic factors of OCS. Some studies conclude that older age, advanced stage at presentation, and suboptimal surgical resection are associated with poor outcome [6,14]. High Ca-125 levels at presentation are an unfavourable prognostic factor [20]. Most of the prognostic factors are accessed based on histopathological examination of the specimen: epithelial carcinoma grade, myometrial vascular invasion, type of epithelial and sarcomatous components, presence of sarcomatous components, vascular endothelial growth factor (VEGF), vascular endothelial growth factor receptor 3 (VEGFR-3) expression, number of small vessels and stromal sarcoma predominance in the primary tumour. Tumour protein p53 and antigen Ki67 obtained by immunohistochemical staining are also prognostic factors [1,6,14,20,21,22,23,24,25,26]. Studies establish the following most significant prognostic factors correlated with better prognosis: patient’s age, tumour stage at diagnosis, type of surgery, epithelial carcinoma histologic grade, epithelial component histology and type of adjuvant chemotherapy [1,6,14,20,21,22,23,24,25,26]. In conclusion, the prognosis of carcinosarcomas is derived from the epithelial component. Prognostic factors related to the sarcomatous components have no longer any prognostic significance [1,6,14,20,21,22,23,24,25,26]. Most of the prognostic factors in OSC mentioned in the medical literature are summarized in Table 1 [1,6,14,20,21,22,23,24,25,26].

In the present case, the patient had 7 favourable (<65 years, optimal surgical resection, no myometrial vascular invasion, no sarcomatous components outside the ovary, no stromal predominance in the primary tumour, absent p53 overexpression, adjuvant platinum-based therapy), and 6 unfavourable prognostic factors (advanced initial tumour stage, high-grade serous carcinoma, serous epithelial components, presence of sarcomatous components in the primary tumour > 25%, heterologous sarcoma components, a high number of small vessels). Tests for Ca-125 levels, Ki67 reactivity, VEGF, and VEGFR-3 expression were not performed.

The treatment suggestion of MMMT is based mainly on small retrospective studies or extrapolations from research on epithelial ovarian cancer or uterine carcinosarcoma [27]. The initial therapy of OCS is optimal surgical cytoreduction including total abdominal hysterectomy, bilateral adnexectomy, omentectomy, selective pelvic and para-aortic lymph node dissection to the left renal vein, and tumour debulking [2]. Systematic lymphadenectomy for the advanced stage is not recommended [28]. However, the definition of optimal cytoreduction has changed over the years. Currently, optimal cytoreduction is defined as no macroscopic residual disease [29]. Although the role of cytoreduction in ovarian carcinoma is well established, some studies have reported no benefit of cytoreduction in OCS [4,30,31]. Duska et al. reported that optimal cytoreduction increased the disease-free interval, but not the overall survival. In the majority of studies, optimal surgery is correlated with better progression-free and overall survival in carcinosarcoma, and complete cytoreduction should be the goal of surgical treatment [11,32,33]. Fertility-sparing surgery in patients with OCS is not recommended regardless of age or stage. However, if the patient chooses fertility-sparing surgery, a hysterectomy is recommended after childbearing [34].

Establishing the most effective adjuvant chemotherapeutic regimen in OCS remains elusive [2,35]. Platinum-based chemotherapy is the most commonly used adjuvant treatment as it has been found to be superior to no platinum-based regimes [1,6,14,28]. In a study, the authors reported that doxorubicin was inadequate as single-agent chemotherapy: it was shown to be insufficient, and a poor response rate and high toxicity were observed [4,36]. Different platinum-based regimes have been used over the years. In a study of 31 patients with OCS, the outcomes between patients treated with carboplatin/paclitaxel compared to those treated with ifosfamide/paclitaxel have been compared. The authors concluded that median progression free-survival (PFS) was longer in patients treated with carboplatin/paclitaxel, whereas overall survival (OS) was similar for all treatment groups [37]. Another study of 31 patients with OCS compared ifosfamide/cisplatin vs. carboplatin/taxol. PFS and OS showed improvement with the use of ifosfamide/cisplatin [33]. Heinzelmann-Schwarz et al. reported that in endometrial carcinosarcoma platinum/anthracycline or ifosfamide regimens were associated with better outcomes than platinum/taxanes regimens so that these findings might be applied to OCS [38]. Yalcin et al. compared the prognosis of patients with OCS to patients with ovarian high-grade serous carcinoma treated in the same manner: optimal cytoreductive surgery followed by platinum plus taxane combination chemotherapy. They stated that the therapeutic approaches could be identical for patients with either diagnosis as PFS and OS were similar for patients with OCS and ovarian high-grade serous carcinoma [39]. However, Brown et al. reported for 65 patients with MMMT and 746 patients with serous ovarian adenocarcinoma selected as a comparison group. They concluded that OCS had an inadequate response to platinum-based chemotherapy, and worse progression-free and cause-specific survival compared to serous ovarian adenocarcinoma [40]. Loizzi et al. treated 13 patients with OCS and compared adjuvant chemotherapy based on the combination of cisplatin, epirubicin and ifosfamide (PEI) or taxol and carboplatin (TAX-CBDCA). They concluded that the median OS was not statistically different between the PEI and TAX-CBDCA regimen groups. However, higher toxicity has been observed in the PEI group [41]. 

Currently, the National Comprehensive Cancer Network (NCCN) guidelines recommend that paclitaxel 175 mg/m^2^/carboplatin (three to nine cycles) is the preferred adjuvant chemotherapy regimen for OCS stage I. Other recommended regimens are carboplatin/liposomal doxorubicin or docetaxel/carboplatin. In certain circumstances carboplatin/ifosfamide, cisplatin/ifosfamide, and paclitaxel/ifosfamide might be used [34]. In stage II–IV, paclitaxel/carboplatin/bevacizumab + maintenance bevacizumab, are added as first-line treatment. Paclitaxel weekly/carboplatin weekly and paclitaxel weekly/carboplatin q 3 weeks are additional second-line treatment regimens. NCCN stated that data on primary systemic therapy regimes for OCS are limited [34].

Neoadjuvant chemotherapy should be considered if optimal debulking cannot be achieved, or for patients who are poor candidates for surgery due to a high risk of operative morbidity or extra-abdominal disease [28]. Intravenous taxane/carboplatin and liposomal doxorubicin/carboplatin regimens are recommended [34]. Nizam et al. reported on two patients who underwent neoadjuvant chemotherapy with 3 cycles of carboplatin/taxol and optimal cytoreductive surgery in one patient, and suboptimal cytoreductive surgery in the other one, respectively [42]. Hyperthermic intraperitoneal chemotherapy (HIPEC) is under consideration, as it is an acceptable treatment option for ovarian cancer, but there are no published data specifically for carcinosarcoma. The combination of intravenous and intraperitoneal chemotherapy is possible [1,3].

The necessity for adjuvant radiotherapy (external beam irradiation and/or vaginal brachytherapy) has not shown any overall survival benefit but has been reported to decrease local recurrence [3,7]. Although there is no evidence against the beneficial effect of radiotherapy, it may be used for patients with chemotherapy-refractory, recurrent or persistent disease that is restricted to the pelvis [12,43]. Nizam et al. reported on two patients with OCS treated with adjuvant radiation therapy. One patient had axillary metastases and the other one axillary recurrence after optimal debulking surgery. They are currently alive [42]. 

Geisler et al. examined nine cases of OCS and found receptors for oestrogen in six of the cases and no receptor for progesterone [44]. In another study of uterine carcinosarcomas, receptors for oestrogen and progesterone were found in four of 11 cases [45]. These findings could suggest the possibility of hormone therapy as a potential therapeutic alternative. 

Given the rarity of the disease, there are limited data regarding the relevance of targeted therapies. Overexpression of COX-2, HER-2/neu, C-kit in carcinosarcomas has been reported [1]. Consequently, agents like imatinib (targeting C-kit) and tyrosine kinase inhibitor (TKI) neratinib (targeting HER-2/neu) may provide an opportunity for targeted therapies [1]. A recent case study reported a significant tumour response in a patient with OCS to the poly (ADP-ribose) polymerase inhibitor Olaparib [42,46].

In the present case, the patient underwent optimal cytoreductive surgery with no macroscopically residual disease and started first-line platinum-based adjuvant chemotherapy paclitaxel/carboplatin/bevacizumab.

The treatment of OCS is summarized in Figure 8 [27,28,29,30,31,32,33,34,35,36,37,38,39,40,41,42,43,44,45,46].

In the present case, optimal cytoreduction was performed. Metastatic para-aortic lymph nodes were the only extrapelvic tumour dissemination. Ligation of the inferior mesenteric artery (IMA) could be applied without any clinical consequences. Surgeons ligate IMA either directly at the origin of the IMA from the aorta (high ligation), or at a point just below the origin of the left colic artery (low ligation) [47]. There are two arterial arches between the middle colic artery (branch of the superior mesenteric artery) and the left colic artery (branch of the inferior mesenteric artery), which maximally keep the blood supply to the descending colon after high ligation of the IMA. The two arches are marginal arterial arch (also known as the artery of Drummond), and Riolan’s arch [48]. Although some reports showed that descending colon, ischemia or necrosis might occur after high ligation of the IMA (especially for patients with advanced rectal or sigmoid cancer, cerebrovascular disease, and hypertension), in patients without general disease they are less likely to occur [48]. 

Small bowel obstruction/ileus is a common reason for readmission after optimal cytoreductive surgery for epithelial ovarian cancer, especially after para-aortic lymphadenectomy [49,50]. The treatment of such an obstruction is decompression of the bowel by a nasogastric tube and aggressive fluid resuscitation. If conservative treatment fails for 72 h, the patient undergoes a relaparotomy [49]. In the present case, the reason for postoperative ileus was adhesion between the terminal ileum and the abdominal aorta. To the best of our knowledge, such a complication is reported for the first time here, and it should be considered as a possible complication after para-aortic lymph node resection.

There are many studies suggesting prevention of postoperative ileus [51,52,53,54,55]. Intravenous infusion of lidocaine decreases the risk of postoperative ileus by reducing pain and, therefore, sympathetic stimulation [51]. Preoperative probiotics administrations, COX-2 inhibitors and carbohydrate loading may have possible beneficial effect of postoperative ileus prevention [53]. A study of 707 patients reported that immediate postoperative feeding and bowel stimulation with 30 mL of magnesium hydroxide (milk of magnesia) twice daily is a safe and effective approach to preventing ileus in patients who undergo major gynecologic surgical procedures [54].

Currently, enhanced recovery after surgery (ERAS) protocols were adopted in gynecologic oncology. ERAS was first used in colorectal surgery and later in urology, vascular surgery and gynecology. ERAS reduce the rate of postoperative ileus by series general measures, such as early feeding, avoidance of mechanical bowel preparation, opioid-sparing pain control, utilization of preoperative patient education, optimization of comorbid conditions, euvolemia, normothermia [51,52,55].

The strengths of the present case include: a brief review of OCS pathogenesis, prognosis, histology, and treatment summarized and illustrated in tables and figures. Retroperitoneal para-aortic lymph node metastasis is the only extrapelvic dissemination of OCS. The postoperative complication described has never been reported in the medical literature. The limitations of this case report include the following: levels of Ca125, preoperative computed tomography, clonality studies, patterns, genomic analysis, gene mutations and expressions were not performed.

## 4. Conclusions

OCS is a rare disease with a dismal prognosis. Retroperitoneal, pelvic, and para-aortic lymph nodes should be closely inspected in patients with ovarian tumours. Retroperitoneal para-aortic lymph node metastasis could be the only extrapelvic dissemination of OCS. There are limited data on aetiology, diagnosis, prognostic factors and treatment of OCS. However, studies concluded that prognostic factors and treatment of carcinosarcomas are associated with epithelial components, as carcinosarcomas are carcinomas with epithelial–mesenchymal transition and heterologous differentiation [1,4,6,20,21,22,23,24,28,29,30,31,32,33,34,35,36,37,38,39,40,41,42,43,44,45,46]. Adhesion between the small bowel and the abdominal aorta leading to obstructive ileus is a possible complication after para-aortic lymph node dissection. 

## Figures and Tables

**Figure 1 diagnostics-10-01073-f001:**
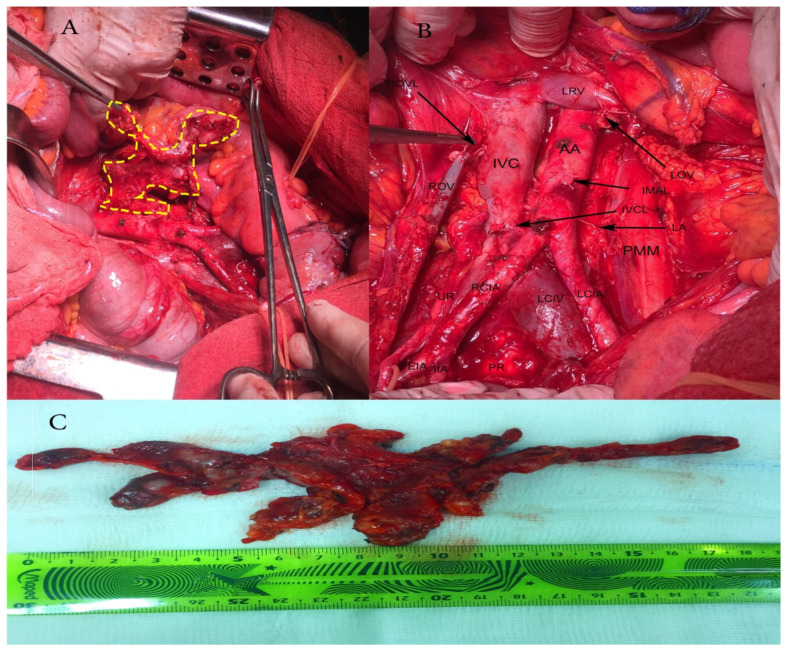
Paraaortic lymphadenectomy. (**A**) Resection of bulky paraaortic lymph nodes suspected for lymph node metastases; (**B**) Paraaortic region after lymphadenectomy; AA—abdominal aorta; IVC—inferior vena cava; LRV—left renal vein; LOV—left ovarian vein, cut and ligated where it drains into the left renal vein; IMAL—inferior mesenteric artery, cut and ligated; IVCL—inferior vena cava severed during pelvic lymphadenectomy, IVC was sutured with 4/0 absorbable suture; LA—lumbar artery; PMM—psoas major muscle; LCIA—left common iliac artery; LCIV—left common iliac vein; PR—promontory; RCIA—right common iliac artery; UR—ureter; EIA—external iliac artery; IIA—internal iliac artery; ROV—right ovarian vein; ROVL—right ovarian vein ligated where it drains in inferior vena cava; (**C**) Paraaortic lymph nodes removed en masse.

**Figure 2 diagnostics-10-01073-f002:**
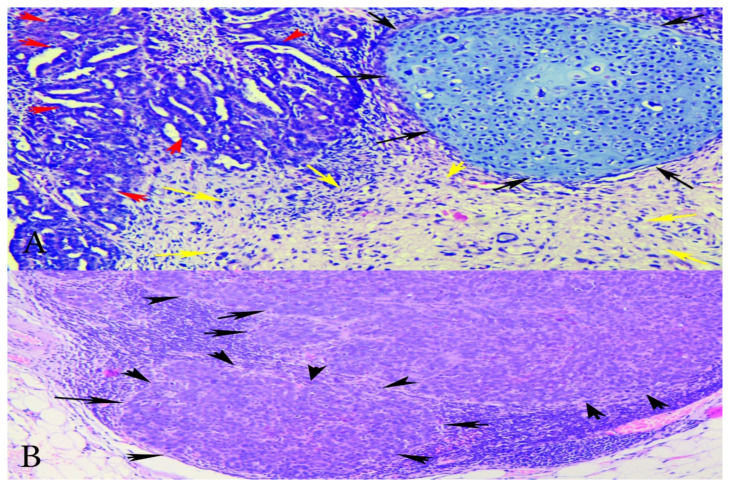
(**A**) Microphotographs of the tumour showing carcinoma and sarcoma components—H&Ex200-carcinomatous with high-grade serous carcinoma and sarcomatous component with stromal and chondroid differentiation. Red arrows—serous epithelial carcinoma; black arrows—chondrosarcoma; yellow arrows—endometrial stromal sarcoma. (**B**) Metastatic lymph node—HEx100 lymph node with metastases of carcinomatous (high-grade serous) component. Black arrows—lymph node metastases from the epithelial component.

**Figure 3 diagnostics-10-01073-f003:**
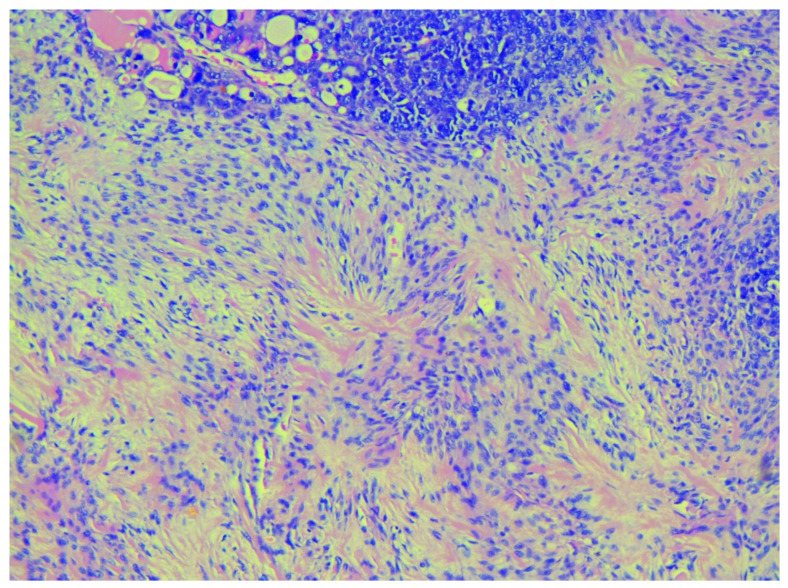
Starburst appearance in the sarcomatous component typical for stromal sarcoma, H&Ex200.

**Figure 4 diagnostics-10-01073-f004:**
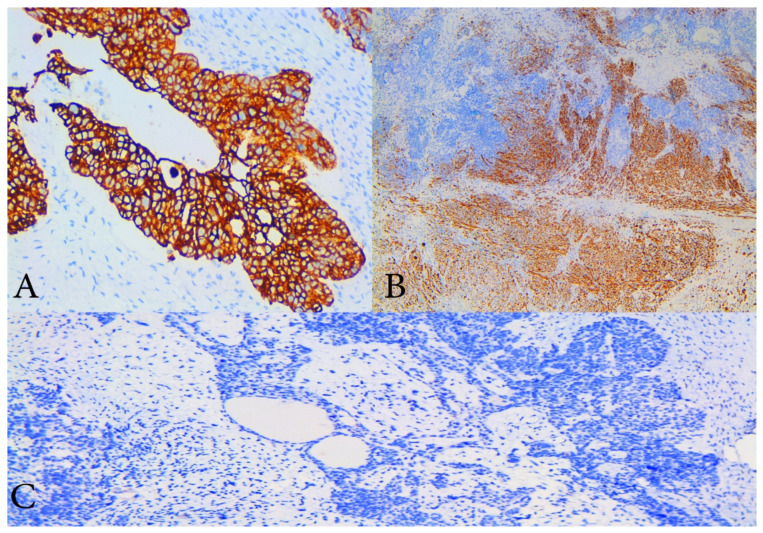
Immunohistochemical staining (IHC). (**A**) IHC with cytokeratin 7—diffuse and strong cytoplasmic positivity in malignant epithelial cells of high- grade serous component of the tumour, ×200 magnification; (**B**) IHC with desmin which shows cytoplasmic positivity of sarcomatous component with smooth muscle differentiation, ×100 magnification; (**C**) IHC with p53 which shows that both components are negative, magnification ×200.

**Figure 5 diagnostics-10-01073-f005:**
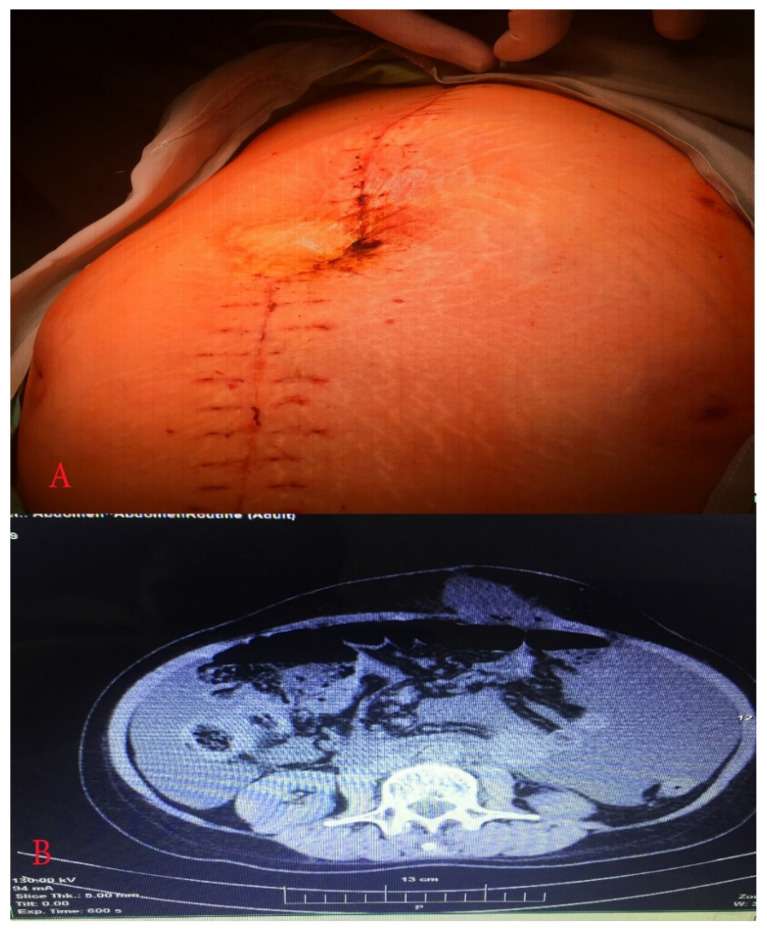
(**A**) Abdomen just before relaparotomy; (**B**) computed tomography showed postoperative ileus and elevated amount of free fluid.

**Figure 6 diagnostics-10-01073-f006:**
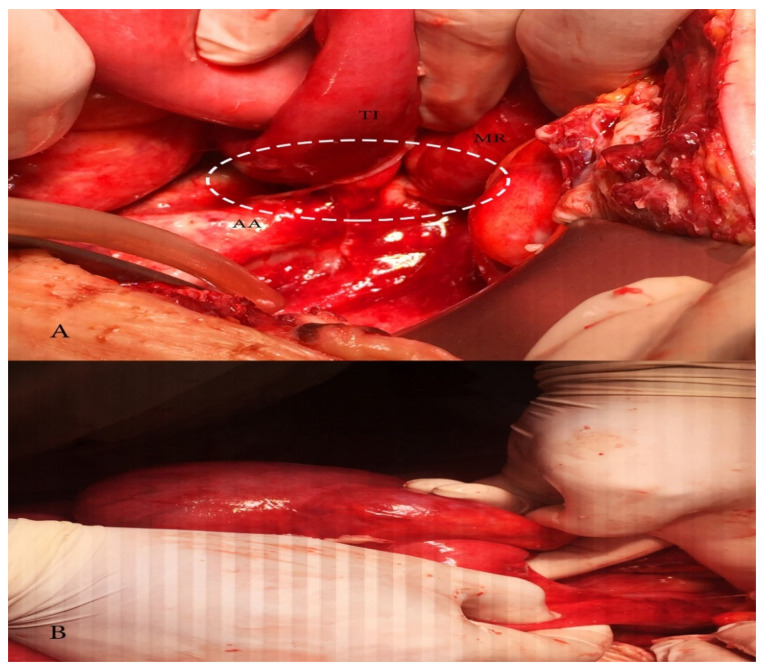
(**A**) Adhesion between terminal ileum and abdominal aorta causing obstructive ileus; MR—mesentery, TI—terminal ileum, AA—abdominal aorta; (**B**) dilatated caecum.

**Figure 7 diagnostics-10-01073-f007:**
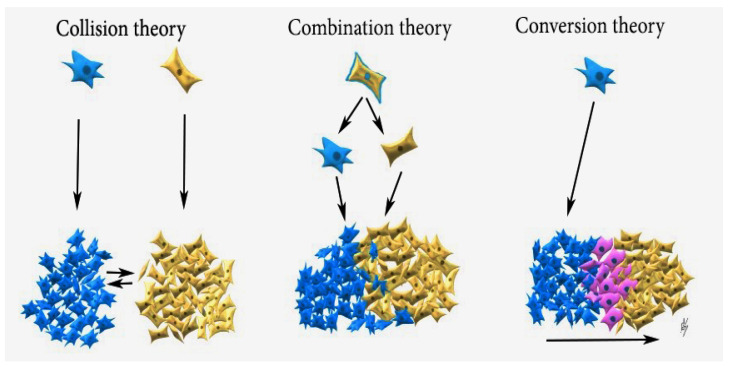
**Collision theory:** Carcinoma (blue cells) and sarcoma (brown cells) are two independent neoplasms, which later merge; **Combination theory:** both components are derived from a single stem cell that undergoes divergent differentiation early in the evolution of the tumour; **Conversion theory:** the sarcomatous element is derived from the carcinoma during tumour evolution (pink cells—epithelial–mesenchymal transition).

**Figure 8 diagnostics-10-01073-f008:**
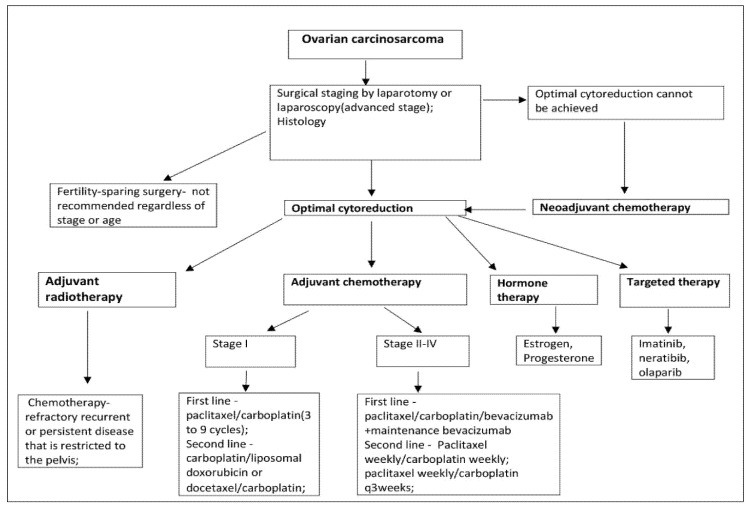
Ovarian carcinosarcoma treatment suggestion.

**Table 1 diagnostics-10-01073-t001:** Prognostic significance of various prognostic factors relevant to ovarian carcinosarcoma.

Prognostic Factors	Favourable	Unfavourable
*Patient’s age*	<65 years	>65 years
*Initial tumour stage*	Early	Advanced
*Surgery*	Optimal surgical resection	Suboptimal surgical resection
*Ca-125 levels at presentation*	Low	High
*Epithelial carcinoma histologic grade*	Low	High
*Myometrial vascular invasion*	No	Yes
*Epithelial components*	Non-serous	Serous
*Sarcomatous components*	Homologous	Heterologous
*Sarcomatous components outside the ovary*	No	Yes
*Presence of sarcomatous components*	<25%	>25%
*Histological stromal sarcoma predominance*	No	Yes
*Number of small vessels in the primary tumour*	Low	High
*VEGF, VEGFR-3 expression*	Normal	Increased
*P53 overexpression*	Absent	Presented
*Ki67 reactivity*	Low	High
*Adjuvant chemotherapy*	Platinum-based regimens	Non-platinum-based regimes

Legend: VEGF—Vascular endothelial growth factor; VEGFR-3—Vascular endothelial growth factor receptor 3; P53—protein 53; Ki67—antigen Ki67.

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
