# Peer review of "Ovarian Carcinosarcoma with Retroperitoneal Para-Aortic Lymph Node Dissemination Followed by an Unusual Postoperative Complication: A Case Report with a Brief Literature Review"

_diagnostics, 2020, doi:10.3390/diagnostics10121073_

Round 1

Reviewer 1 Report

After detailed review the content, this case report should be revised as the following suggestions.

  1. Carcinosarcoma of the ovary is not so rare. Besides, any image to prove the adhesion between the area of paraaortic lymph node dissection and intestine? Such as CT scan? Is there any transition zone between obstruction and non-obstruction area in CT scan or surgical finding?
  2. Besides, the authors did not mention about how to avoid this complication. Close the peritoneum after primary surgery? The authors did not mention about if the peritoneum is close or not in the first time of laparotomy.
  3. The authors reviewed a lot of references about carcinosarcoma. However, the aim of this case report became ambiguous. Primary aim of this case report should be related to describe the mechanism of the complication and to prevent this rare complication, not review of carcinosarcoma of the ovary. If the authors wound like to review the characteristics of carcinosarcoma, the content should be more short and concise, not so lenthy.
  4. The stage of this case was not mentioned.
  5. The management and prevention of complication should be moved to first paragraph of the discussion section.
  6. The treatment flow chart should be mentioned as "suggestion". How about the role of intraperitoneal chemotherapy. The author should cite the following paper ( DOI: 10.3390/ijerph17103603) if no bevacizumab. 

Author Response

Reviewer 1.

  1. Carcinosarcoma of the ovary is not so rare.

           Author’s Reply: As the majority of authors worldwide, we think that Carcinosarcoma of the ovary is a rare disease. Moreover, the word “rare” for ovarian carcinosarcoma exists in every cited article in our manuscript.

 Besides, any image to prove the adhesion between the area of paraaortic lymph node dissection and intestine? Such as CT scan? Is there any transition zone between obstruction and non-obstruction area in CT scan or surgical finding?

Author’s Reply:  We agree with the reviewer. We performed a CT scan. A picture of the CT scan, showed a high amount of free fluid and dilatated small bowels and caecum. We discussed the CS- scan with radiologist and they said that it is impossible to observe the transition zone between obstruction and non- obstruction area. They concluded that the patient had postoperative ileus. That’s why we used intraoperative findings, which are incorporated in the article. We used CT –scan photo to show the bowels dilatation and high amount of free fluid. Intraoperative findings more preciously revealed the adhesion and the transition zone.  Even caecum dilatation is showed.

2.Besides, the authors did not mention about how to avoid this complication. Close the peritoneum after primary surgery? The authors did not mention about if the peritoneum is close or not in the first time of laparotomy.

Author’s Reply: Peritoneum closure is impossible after such an extend paraaortic lymphadenectomy. We agree with the reviewer and mentioned other methods for preventing the postoperative complication.

The next text was inserted:

There are many studies reporting prevention of postoperative ileus.

Intravenous infusion of lidocaine decreases the risk of postoperative ileus by reducing pain and therefore sympathetic stimulation. Preoperative probiotics administrations, COX-2 inhibitors and carbohydrate loading may have possible beneficial effect of postoperative ileus prevention.

 A study of 707 patients reported that an immediate postoperative feeding and bowel stimulation with 30 mL of magnesium hydroxide (milk of magnesia) twice daily is a safe and effective approach to preventing ileus in patients who undergo major gynecologic surgical procedures .

Currently, enhanced recovery after surgery (ERAS) protocols were adopted in gynecologic oncology. ERAS was first used in colorectal surgery and later adapted to urology, vascular surgery and gynecology. ERAS reduce the rate of postoperative ileus by series general measures, such as early feeding, avoidance of mechanical bowel preparation, opioid-sparing pain control, utilization of preoperative patient education, optimization of comorbid conditions, euvolemia, normothermia.

References 50 – 55 were incorporated.

3.The authors reviewed a lot of references about carcinosarcoma. However, the aim of this case report became ambiguous. Primary aim of this case report should be related to describe the mechanism of the complication and to prevent this rare complication, not review of carcinosarcoma of the ovary. If the authors wound like to review the characteristics of carcinosarcoma, the content should be more short and concise, not so lengthy.

Author’s Reply:  Firstly, the purpose of the article was to introduce a complication that had never been reported. Secondly, we decided to make a brief review of the literature about this rear disease. That’s why we chose Journal Diagnostics, as it is an exceptional journal and there are no references limits.

As the majority of authors, we think that ovarian carcinosarcoma is a rare disease and we decided to summarize most of the important information. If we exclude some of the references, it won’t be a case report with literature review, but only a case report. We think that when the word review is in the Title, at least 50 references should be cited.

4.The stage of this case was not mentioned.

Author’s Reply: It was mentioned before .  FIGO and TNM staging. Even a references have been used.

The patient was diagnosed with surgical stage IIIA1 according to the latest International Federation of Gynecology and Obstetrics (FIGO) classification, and pT2aN1bMx, according to the TNM classification [10].

  1. The management and prevention of complication should be moved to first paragraph of the discussion section.

Author’s Reply: As in the majority of articles, we believe that the best structure of a discussion is as it follows - introduction, epidemiology, ethilogy, clinical presentations, pathophysiology, pathology, risk factors and then Treatment (management, complications and prevention). We do not think that management and prevention should be moved, especially as a first paragraph of the discussion.

6.The treatment flow chart should be mentioned as "suggestion".

Author’s Reply: We agree with the reviewer. The recommended change has been incorporated.

 How about the role of intraperitoneal chemotherapy. The author should cite the following paper ( DOI: 10.3390/ijerph17103603) if no bevacizumab.

Author’s Reply: We stated that intraperitoneal chemotherapy could be a method of treatment, but it is under consideration, as it is an acceptable treatment option for ovarian cancer, but there are no published data specifically for carcinosarcoma. Firstly, the article you wanted to cite is only for ovarian carcinoma. Although, it is stated that carcinosarcomas are actually carcinomas, it can not be used for carcinosarcomas. Secondly, we cited National Comprehensive Cancer Network guidelines. They recommend Bevacizumab. We do not think that an article including 50 patients is superior to NCCN guidelines (especially a research article, which is not the topic of our discussion – ovarian carcinoma, not Carcinosarcoma).

We sorry if our article will be rejected for not citing this paper.

Thank you very much for reviewing the article.

Reviewer 2 Report

minor language revision is needed

Author Response

Reviewer 2. Minor language revision is needed.

Author’s Reply : A native English speaker carefully revised the whole manuscript.

Thank you very much for reviewing and approving the article. I greatly appreciate you taking the time to review it!

Reviewer 3 Report

This case report of an ovarian carcinosarcoma provides a nice overview of this rare type of ovarian malignancy.  While this report is not novel, it does provide an up to date review of treatment options.  The post surgical complication reported is not totally unexpected given the extent of the lymphadenectomy required.  It would be helpful if the authors could comment on ways to prevent this in the future.  Moderate English editing and spell checking is needed.

Author Response

Reviewer 3. This case report of an ovarian carcinosarcoma provides a nice overview of this rare type of ovarian malignancy.  While this report is not novel, it does provide an up to date review of treatment options. 

  1. The post-surgical complication reported is not totally unexpected given the extent of the lymphadenectomy required. It would be helpful if the authors could comment on ways to prevent this in the future. 

Author’s Reply: We agree with the reviewer.

The next text was inserted:

There are many studies suggesting prevention of postoperative ileus.

Intravenous infusion of lidocaine decreases the risk of postoperative ileus by reducing pain and therefore sympathetic stimulation. Preoperative probiotics administrations, COX-2 inhibitors and carbohydrate loading may have possible beneficial effect of postoperative ileus prevention.

 A study of 707 patients reported that an immediate postoperative feeding and bowel stimulation with 30 mL of magnesium hydroxide (milk of magnesia) twice daily is a safe and effective approach to preventing ileus in patients who undergo major gynecologic surgical procedures .

Currently, enhanced recovery after surgery (ERAS) protocols were adopted in gynecologic oncology. ERAS was first used in colorectal surgery and later adapted to urology, vascular surgery and gynecology. ERAS reduce the rate of postoperative ileus by series general measures, such as early feeding, avoidance of mechanical bowel preparation, opioid-sparing pain control, utilization of preoperative patient education, optimization of comorbid conditions, euvolemia, and normothermia.

References 50 – 55 were incorporated.

  1. Moderate English editing and spell checking is needed.

Author’s Reply:  A native English speaker carefully revised the whole manuscript.

Thank you very much for reviewing the article. I greatly appreciate you taking the time to review it!